# Sleeping with time in mind? A literature review and a proposal for a screening questionnaire on self-awakening

Laura Verga[1,2]*, Giada D'Este[3,4], Sara Cassani[3,4], Caterina Leitner[3,4], Sonja A. Kotz[2], Luigi Ferini-Strambi[3,4], Andrea Galbiati[3,4]*

1 Research Group Comparative Bioacoustics, Max Planck Institute for Psycholinguistics, Nijmegen, The Netherlands, 2 Department of Neuropsychology and Psychopharmacology, Maastricht University, Maastricht, The Netherlands, 3 Department of Clinical Neurosciences and Department of Neurology, Sleep Disorders Center, San Raffaele Scientific Institute, Milan, Italy, 4 Faculty of Psychology, Vita-Salute San Raffaele University, Milan, Italy

* laura.verga@maastrichtuniversity.nl, laura.verga@mpi.nl (LV); andrea.galbiati.unisr@gmail.com (AG)

## Abstract

Some people report being able to spontaneously "time" the end of their sleep. This ability to self-awaken challenges the idea of sleep as a passive cognitive state. Yet, current evidence on this phenomenon is limited, partly because of the varied definitions of self-awakening and experimental approaches used to study it. Here, we provide a review of the literature on self-awakening. Our aim is to i) contextualise the phenomenon, ii) propose an operating definition, and iii) summarise the scientific approaches used so far. The literature review identified 17 studies on self-awakening. Most of them adopted an objective sleep evaluation (76%), targeted nocturnal sleep (76%), and used a single criterion to define the success of awakening (82%); for most studies, this corresponded to awakening occurring in a time window of 30 minutes around the expected awakening time. Out of 715 total participants, 125 (17%) reported to be self-awakeners, with an average age of 23.24 years and a slight predominance of males compared to females. These results reveal self-awakening as a relatively rare phenomenon. To facilitate the study of self-awakening, and based on the results of the literature review, we propose a quick paper-and-pencil screening questionnaire for self-awakeners and provide an initial validation for it. Taken together, the combined results of the literature review and the proposed questionnaire help in characterising a theoretical framework for self-awakenings, while providing a useful tool and empirical suggestions for future experimental studies, which should ideally employ objective measurements.

## Introduction

Human sleep is a complex physiological function. Behaviourally, it alternates with waking state and is characterised by an increased threshold to sensory input, reduction of motor output, typical changes in central and peripheral physiology, and diminished conscious awareness [1]. Importantly, from a psychological standpoint, sleep plays a crucial role in the regulation of

**Data Availability Statement:** Data and code are available in OSF (https://osf.io/kaunw/). All the material is public and freely available for download and use.

**Funding:** The author(s) received no specific funding for this work.

**Competing interests:** The authors have declared that no competing interests exist.

daytime cognitive functions like memory and emotion [2–4]. Moreover, other phenomena (e.g., lucid dreaming) suggest that high order cognitive processing may be active during sleeping. This suggests that awareness may re-emerge during this apparently inactive brain state. For example, lucid dreaming, namely the ability of being aware of dreaming while dreaming, assumes a form of volitional control in sleep [5]. Another understudied experience related to a regained cognitive activity during sleep is self-awakening. In daily life, people use alarm clocks to wake up at a specific time in the morning. Interestingly, some report the ability to awaken without the help of any timekeeper; we refer to this phenomenon as self-awakening. Although the experience is commonly reported, some specifications are warranted to identify and understand the phenomenon (see Fig 1). First, awakening from sleep might be induced by external stimuli but can also occur spontaneously. Second, spontaneous awakening might be subdivided into natural awakening, caused by a "natural" cessation of sleep due to the dissipation of the physiological drive to sleep, or self-awakening due to the willingness to wake up. Third, self-wakening itself needs to be disambiguated: self-awakening might be habitual (i.e., a person waking up every day at the same time), in which case the interaction between circadian rhythms and learned behaviour is the driving factor, or it can be induced by volition [6]. This latter phenomenon represents the topic of interest here.

Self-awakening remains largely understudied; however, the first systematic investigation of self-awakening dates to the end of the 19th century. In his work entitled "Statistics of Unconscious Cerebration", Charles Manning Child investigated this ability in a survey by asking the following question: "Can you wake precisely at a given hour determined upon before going to sleep, without waking up many times before the appointed time?". The author found that 59% of the participants (200 in total) reported waking up at a predetermined time without disturbed sleep [7]. Yet, the first empirical study on self-awakening was conducted by Nicolae Vaschide at the beginning of 20th century [8]. Participants, who habitually woke up at 8 a.m., were asked to wake up at three different times during the night (at 3, 5, and 6 a.m.) without the help of any external cue. Interestingly, the mean error (i.e., the discrepancy between target time and actual wake-up time) was dependent on a specific wake-up time. Participants woke up more accurately during the last part of the night. This might reflect the influence of slow

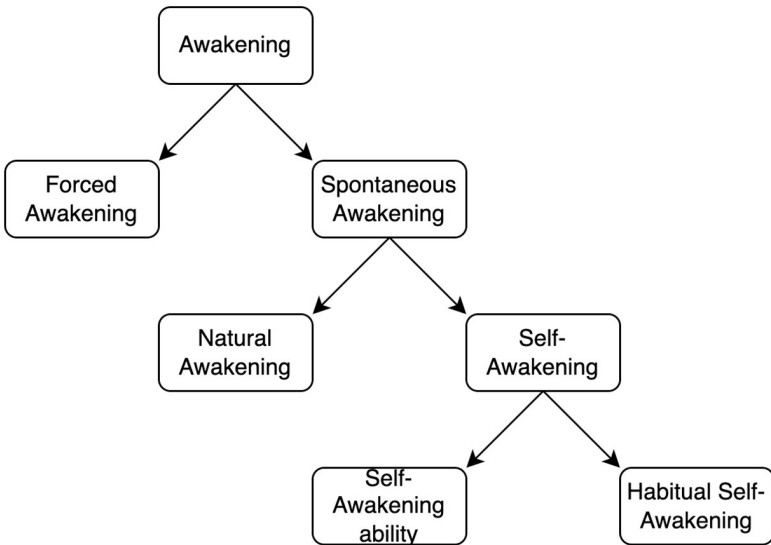

**Fig 1. Schematic representation of a taxonomy for awakenings.**

wave sleep, which predominates the first part of the night and is characterised by minimal consciousness [9]. Accordingly, one possible explanation for the phenomenon of self-awakening might involve the subjects' ability to disengage from slow wave sleep [10].

The study of self-awakening is not only interesting, but it could also inform on the presence of specific cognitive processes even at minimal or absent levels of consciousness. There are at least three cognitive operations a sleeping person must manage to awaken at the right point in time during sleep: first, one must encode and store the target time, which must be readily accessible; second, the time elapsed during sleep should be constantly estimated and compared to the target time; third, one must regain volitional control during sleep to awake. All these operations must be carried out while the sleeping subject is in a state (i.e., sleep) typically characterised by minimal levels of consciousness. This suggests that some cognitive processes might be active during sleep in self-awakeners. However, before addressing this issue, it is necessary to properly characterise the concept of self-awakening and to streamline the methods to study this phenomenon. Up to now, self-awakening has been defined and studied using different subjective and objective methodologies, including surveys, sleep diaries, actigraphy, and polysomnography. The aim of this article is to review the scientific literature on self-awakening intended as an ability, in line with the taxonomy reported in Fig 1. Furthermore, based on the literature review, we propose a quick paper-and-pencil questionnaire that could provide a first screening tool for non-clinical self-awakeners.

## Methods

### Literature review

This review follows the procedure and guidelines of the Preferred reporting items for reviews and meta-analyses (PRISMA) statement, elaboration, and explanation [11–14].

**Search method.** Two researchers (GD, SC) independently searched the relevant literature up to October 6th, 2022, in the following databases: Pubmed, Web of Science, Embase, Scopus, PsycInfo, and ProQuest. Search operators (e.g., AND; the proximity operator NEAR/0 was employed to find records where the search terms joined by the operator were adjacent) were used to combine the following the key-terms: self-awakening, self NEAR/0 awakening, forced-awakening, forced NEAR/0 awakening, self-awakening AND nap, self-awakening AND daily sleep, self NEAR/0 awakening AND nap, self NEAR/0 awakening AND daily sleep. Subsequently, other key-terms were searched in the same databases: ability to wake from sleep, ability to awaken from sleep, ability to awake from sleep, awakening at a pre-set time, awakening at a preselected time, wakening at a preselected time, waking from sleep at a preselected time, awakening from sleep at a pre-set time, (time NEAR/0 perception) AND (REM NEAR/0 sleep). The relevant key terms could appear anywhere in a manuscript (e.g., title, abstract, body). Peer-reviewed and non-peer reviewed materials (e.g., conference abstracts appearing in Special Issues) were considered to avoid publication bias; yet only full manuscripts were retrieved from the database search. We allowed the search to include manuscripts written in languages other than English; however, only English studies were retained in the final set of suitable papers based on the criteria outlined in the following section.

**Selection of studies.** Abstract were first scanned and evaluated for eligibility. The reference list of the selected papers was carefully inspected to identify other suitable manuscripts. Manuscripts were selected if they i) contained experimental studies conducted either in a sleep laboratory or at home (sleep diaries, questionnaires) and/or ii) evaluated spontaneous awakenings during daytime or night-time sleep; they were excluded if they were i) non-experimental (e.g., observational) or ii) in case of abstracts, they coincided with published journal articles.

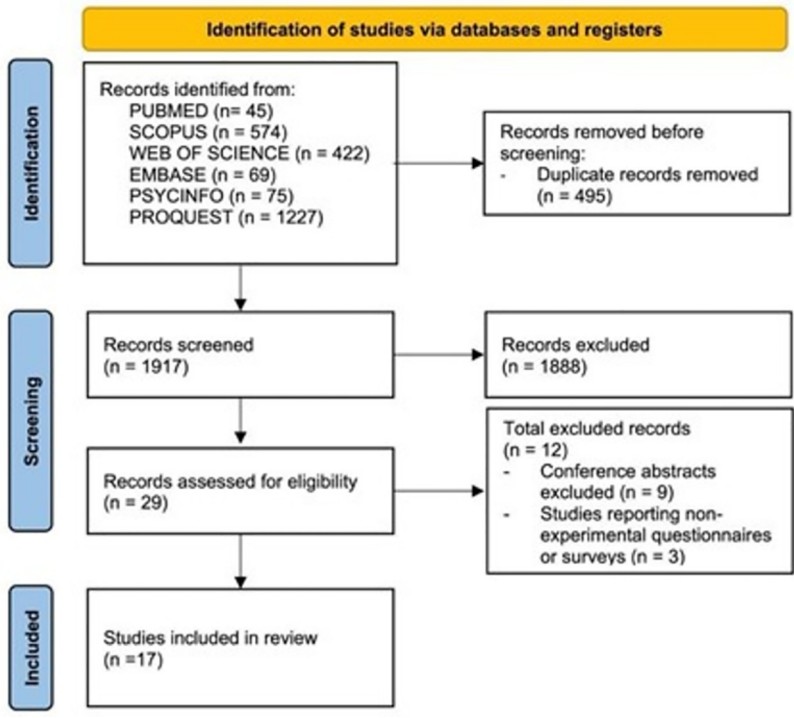

**Fig 2. PRISMA flow diagram summarizing the selection procedure.**

Disagreements between the two independent evaluators were reviewed and solved by internal discussion. The selection procedure is summarized in the PRISMA flowchart (Fig 2).

**Quality evaluation.** The quality of the studies selected for the literature review was assessed using the Critical Appraisal Skills Program (CASP) for cohort studies tool. This checklist aims at reliably evaluating each study in a systematic way. Each of the 12 questions assesses a specific aspect of the study, namely: clarity of the research question (Question 1), sample recruitment (Question 2), classification bias (Question 3, 4), confounding factors (Question 5a, 5b), follow-up characteristics (Question 6a, 6b), preciseness, validity, and applicability of the results (Question 7, Question 8, Question 9, Question 10), fitness of the results within the literature (Question 11), implications of the study (Question 12).

The evaluators independently rated the quality of each study; results of this assessment were compared, and disagreements were solved by internal discussion. Each study could obtain a maximum of 14 points; the average quality score for the selected studies was 10.71 (median = 11; range = 9–13). Since all studies scored above 50% of the maximum result, none was excluded from subsequent analyses based on the quality ratings. Cohen's Kappa statistic was calculated and the overall agreement between two raters was moderated (k = 0.58).

## Self-Awakening Questionnaire (SAQ)

A screening questionnaire was created by LV and AG to target and identify possible non-clinical self-awakeners. The questionnaire included 5 questions based on the reviewed literature on self-awakeners and was translated into both English (Table 1) and Italian (see S1 File). SAQ was originally created in Italian by the first and last authors (LV and AG) and subsequently translated into English. The English version was checked by two native speakers of American

**Table 1. Self-Awakening Questionnaire (English version).** The questionnaire consists of five questions aiming at probing the ability of a subject to wake up spontaneously (Question 1), and accurately (Question 4), independently from habit or circadian preference (Question 2, Question 5), or underlying conditions (e.g., stress, anxiety; Question 3).

| Self-Awakening Questionnaire (SAQ) | |
|---|---|
| Aim | Under some circumstances—for example on the night before a test or an important meeting at work—some people are able to wake up spontaneously at a specific time, anticipating the sound of their alarm clock or even without the need to set one. The following questions aim at characterising this behaviour. |
| Instructions | Read each question carefully. |
| | Select the response that better defines your personal experience. |
| | Please answer **ALL** questions. |
| Q1 | Have you ever spontaneously woken up at a desired time, without the help of an alarm clock or other external devices?<br>• yes<br>• no |
| Q2 | If yes, how frequently?<br>• Every day<br>• At least once a week<br>• Only on nights before important events (Which ones? Please specify) |
| Q3 | When you experience these spontaneous awakenings, do you wake up more than once throughout the night?<br>• Yes<br>• No, just in the morning before the alarm clock |
| Q4 | On average, how many minutes before the desired time do you wake up?<br>• 5 minutes before or less<br>• Between 5 and 15 minutes<br>• Between 15 and 30 minutes<br>• More than 30 minutes (please specify . . . . . . . . . . . . . . . . . . . . .) |
| Q5 | In your experience, these spontaneous awakenings are more likely to occur:<br>• When you have to wake up at the usual time (e.g., 7 a.m. on a working/school day)<br>• When you have to wake up at an unusual time (e.g., 3 a.m.) |

English and by several colleagues using English daily as their main working language. The final version was then re-translated in Italian by the other co-authors.

The five questions aimed at characterising a self-awakener as a person capable of spontaneously waking up from sleep at a preferred time (Table 1, Question 1 and 2) and accurately (Table 1, Question 4), not simply because of a habit (i.e., not at the usual waking time; Table 1, Question 2 and 5), or due to disrupted sleep caused by stress, anxiety, or other causes (i.e., multiple awakenings during the night; Table 1, Question 3).

The questionnaire was administered to 164 students at Maastricht University (Maastricht, the Netherlands). All students were enrolled in the English track and used English daily as their main study language. Given the paucity of previous work on self-awakening and the exploratory nature of our study, we did not perform an a-priori power analysis to estimate an appropriate sample size. Yet, a number of participants > 150 can be considered a reasonable sample size for most types of analyses, especially for simple statistics as the ones performed in the current study. The study was advertised locally on bulletin boards across campus and on the SONA system of Maastricht University. Interested participants could either contact the experimenters directly via email or could select the study from those available on the SONA system, a participant pool management system used by many universities (https://www.sona-

systems.com/). In SONA, interested students can easily find and enrol in ongoing studies in exchange for university credits or cash. All the participants in the current study were compensated with university credits. Recruitment stopped when no further participants would sign up for the experiment after 2 months since the enrolment of the last participant. Ethical approval was granted by the local ethical committee of Maastricht University, Faculty of Psychology and Neuroscience (agreement nr. OZL_207_04_04_2019). Every subject gave written informed consent to participate in the study.

The questionnaire was complemented by additional tests to probe sleep-related aspects (e.g., sleep quality, insomnia, circadian preference), as well as psychological traits and disorders known to influence sleep (e.g., anxiety, depression, arousal). The following tests were administered: 1) Insomnia Severity Index (ISI [15]), 2) Morningness-Eveningness Questionnaire (MEQ [16]), 3) Pittsburgh Sleep Quality Index (PSQI [17]), 4) Penn State Worry Questionnaire (PSWQ [18]), 5) State-Trait Anxiety Inventory (STAI [19]), 6) Center for Epidemiologic Studies Depression Scale (CES-D Scale [20]), 7) Ruminative Response Scale (RRS [21]), 8) Pre-Sleep Arousal Scale (PSAS [22]). All questionnaires were administered online via Qualtrics (Qualtrics, Provo, UT); total testing time was approximately 30 minutes.

A score from 0 to 3 was assigned to each response given to SAQ, leading to a maximum total score of 13 (see S1 Table in S1 File for details on the score assigned to each question). We set the cut-off to define a self-awakener at 11/13, based on our theoretical definition of self-awakener as well as on the score distribution. More specifically, the combination of the answers and scores needed to identify a self-awakener was as follows:

- Question 1: "yes", 1 POINT

- Question 2: "only on nights before important events", 3 POINTS

- Questions 3: "No, just in the morning before the alarm clock", 3 POINTS

- Question 4: stratified answer, 1, 2, or 3 POINTS

- Question 5: "When you have to wake up at an unusual time", 3 POINTS

Thus, the minimum score needed to be defined as a self-awakener is 11, with some variability granted by Question 4.

Based on this cut-off, we divided participants in two groups: self-awakeners (SA, n = 9) and non-self-awakeners (nSA, n = 144). To evaluate possible differences between these two groups (self-awakeners vs. non-self-awakeners), separate non-parametric Mann-Whitney tests were performed using the outcome of each test of the battery as dependent variable. To account for the difference in sample size between the two groups, we bootstrapped each Mann-Whitney test (1000 repetitions); at each repetition, the nSA sub-group was randomly resampled (with replacement) from the global sample of 144 nSA individuals.

In addition, for the whole group of participants (i.e., without grouping them into SA and nSA), Spearman rank correlation tests were conducted to evaluate possible relationships between the SAQ scores and the answers given to other tests in the battery. Correlations tests were Bonferroni corrected for multiple comparisons (p = .05/number of performed tests).

Lastly, we evaluated the impact of circadian preference, as it has been reported that circadian preference might play a role in self-awakening [6, 23, 24]. Participants were divided into three groups (morning, intermediate, and evening type) based on their MEQ responses. Their responses to each SAQ question were analysed separately by means of a non-parametric ANOVA (Kruskal-Wallis test) to account for differences in data distribution between the three groups (S1 Table in S1 File).

All statistical analyses were conducted in R (version 4.1.2, https://www.R-project.org/ [25]) running in R studio (version 1.3.959, http://www.rstudio.com/ [26]).

## Results

### Literature review

Based on the search criteria, the database research identified 29 papers targeting the topic of self-awakening. 17 of these were published in peer-reviewed journals, 9 appeared as conference abstracts, and 3 reported non-experimental questionnaires or surveys. All retrieved articles and abstract were written in English. Most conference abstract (6 out of 9) coincided with published articles; the remaining 3 abstract reported non-experimental studies. Thus, only the 17 peer-reviewed journal articles were retained in the review (Table 2).

**Study type.** The majority of retained papers (n = 13/17, 76%) included an objective sleep evaluation being either PSG (n = 11/17, 65%), a combination of Electroencephalography (EEG) and Electrooculography (EOG) (n = 1/17, 6%), or actigraphy (n = 1/17, 6%); four papers (24%) included studies conducted either at home (n = 3/17, 18%) or in a sleep lab (n = 1/17, 6%) without objective validation. 13 out of 17 papers (76%) investigated self-awakenings from nocturnal sleep; the remaining 4 papers (24%) focused on diurnal periods of sleep (naps). Most papers (n = 13/17, 76%) described a single study, while four papers (24%) reported two studies; of these, three used the first study to select or confirm self-awakeners to be tested in the second study and only one tested a different sample in the second study.

**Participants' characteristics.** All experiments combined tested 715 participants, of whom 299 were males (42%) and 392 were females (55%); for 24 participants (3%), this information was not specified. 14 papers reported the age of participants: 10 reported the average age (23.24 years, standard deviation (SD) = 6.24) in one or both studies described therein, yet, only 9 of these also reported the SD (average reported SD = 2.81 years); 4 papers reported an age range (20–45, single study; 21–30, single study; 15–32, single study; 21–84 for study 1 and 19–62 for study 2); 3 papers did not report any information on participants' age.

**Self-awakeners: Definition and details.** Most papers (n = 16/17, 94%) reported a time criterion to define a successful self-awakener (i.e., the spontaneous awakening had to occur within a specific time window around the expected time to represent a hit), while one paper (6%) did not report any criterion. Studies reporting a criterion employed a single criterion (n = 14/17 papers, 82%) or, more rarely, multiple criteria (n = 2/17 papers, 12%). Of those employing a single criterion, the most employed criterion defined a self-awakener as a person who spontaneously wakes up within 30 minutes of a defined wake-up time (7 papers); other criteria targeted a time of 10 minutes (2 papers), 15 minutes (1 paper), or 5 minutes (4 papers) around the desired waking time. This narrow time criterion (5 minutes) was used by all the studies targeting diurnal naps of short duration (typically 20 minutes). Papers adopting multiple criteria opted for four-time windows around an expected time (10, 20, 30, and 40 minutes), or two-time windows (10 or 15 minutes).

11 papers reported the number of self-awakeners identified based on their criteria, leading to a total of 125 self-awakeners out of 715 total subjects (17%); 4 papers only reported the number of times successful self-awakening occurred, but did not define self-awakeners based on this result; 2 papers reported neither. The percentage of self-awakeners on the total number of participants in each study, either explicitly reported in the papers or calculated based on the available data, ranged from a minimum of 10% to a maximum of 90% (average percentage = 52%). This percentage is much higher than the 17% based on our calculations.

Eight papers reported the number of males (n = 37) and females (n = 35) among their self-awakeners. These numbers indicate a no difference in the gender of self-awakeners when

**Table 2. Results of the literature review.** "Experiment type": location of the experiment and type of objective sleep validation (PSG = polysomnography, EEG = electroencephalography; EOG = electrooculography; Act = Actigraphy); in case of multiple studies within a paper, S1 indicates the first and S2 the second study. "Sleep type": nocturnal or diurnal sleep. "Nr. Participants": total experimental sample size. "Age (SD)": age range or average for the whole participant group and corresponding standard deviations (SD) when available. "Successful criterion": criterion used to define a successful self-awakener, usually a time window around the expected awakening time (absolute value in minutes). "Nr. SA": number of self-awakeners explicitly reported in the study or whether the study investigated successful self-awakening events. "% SA/total": percentage of self-awakeners on the study sample size either reported in the manuscript or calculated based on the available data. "SA age (SD)": age of self-awakeners and its standard deviation. Additional notes, symbols, and abbreviations: *15 participants were recruited but only 12 underwent EEG. ** depending on the questionnaire question and concerning the ability to wake-up with or without an alarm; *** five participants follow within the second criterion and are considered "moderately successful"; § = studies designed to select only self-awakeners; # = 9 self-awakeners were identified but data of one were lost, leaving 8 participants; ## = PSG failed for one participant, leaving 8 participants (6 F; mean age = 21.8 ±.6 years).

| | Article | Experiment type | Sleep type | Nr. Participants (males / females) | Age (SD) | Successful criterion \|minutes\| | Nr. SA (males/ females) | % SA/ total | SA age (SD) |
|---|---|---|---|---|---|---|---|---|---|
| 1 | Tart, 1970 [27] | S1: home exp | nocturnal | 10 | | \|10\| | events | | |
| | | S2: sleep lab (PSG) | nocturnal | 3 (2 /1) | 18 (—) | \|10\| | events | | |
| 2 | Zung & Wilson, 1971 [28] | sleep lab | nocturnal | 22 (4 / 18) | 20–45 | \|10\| | events | | |
| 3 | Lavie et al., 1979 [29] | S1: sleep lab (PSG) | nocturnal | 7 (6 / 1) | 21–30 | \|10\| \|20\| \|30\| \|40\| | events | | |
| | | S2: sleep lab (PSG) | nocturnal | 2 | | \|20\| | events | | |
| 4 | Bell, 1980 [30] | home exp | nocturnal | 38 (20 /18) | | | events | | |
| 5 | Zepelin, 1986 [31] | S1: sleep lab (EEG, EOG) | nocturnal | 15 * (7 / 8) | 15–32 | \|15\| | 11 | 73 | |
| | | S2: sleep lab (EEG, EOG) | nocturnal | 2 | | | events | | |
| 6 | Hawkins, 1989 [32] | home exp | nocturnal | 84 (31 / 53) | 22.7 (6.7) | \|30\| | 22 | 26 | |
| 7 | Hawkins & Shaw, 1990 [33] | home exp | nocturnal | 146 (38 / 108) | 22.9 (6.2) | \|30\| | 15 | 10 | |
| 8 | Moorcroft et al., 1997 [34] | S1: telephone survey | nocturnal | 269 (129 /140) | 21–84 | | | 23, 29, 24 ** | |
| | | S2: home exp (Act) | nocturnal | 15 (6 / 9) | 19–62 | \|10\| \|15\| | 5 5 *** | 33 | |
| 9 | Kaida, Nakano, et al., 2003 § [35] | sleep lab (PSG) | diurnal | 11 (4 / 7) | 21.7 (1.25) | \|5\| | 9 (3 / 6) | | 21.6 (1.24) |
| 10 | Kaida, Nittono, et al., 2003 [36] | sleep lab (PSG) | diurnal | 14 (6 / 8) | 21.3 (1.3) | \|5\| | 10 (4 / 6) | 71 | |
| 11 | Kaida et al., 2005 § [37] | home exp (sleep diary, PSG) | diurnal | 10 | | \|5\| | 9 (7 / 2) | | 74.1 (5.01) |
| 12 | Kaida et al., 2006 § [38] | home exp (sleep diary, PSG) | diurnal | | | \|5\| | 9 (7 / 2) | | 74.1 (5.0) |
| 13 | Ikeda & Hayashi, 2008 § [39] | sleep lab (PSG) | nocturnal | 10 (3 / 7) | 21.7 (.06) | \|30\| | 8 # (2 / 6) | | 21.8 (0.7) |
| 14 | Matsuura & Hayashi, 2009 [40] | sleep lab (PSG) | nocturnal | 17 (10 / 7) | 19.7 (1.55) | \|30\| | 11 (5 / 6) | 65 | 19.0 (1.73) |
| 15 | Ikeda & Hayashi, 2010 [41] | sleep lab (PSG) | nocturnal | 10 (3 / 7) | 21.8 (.6) | \|30\| | 9 ## (2 / 7) | 90 | 21.8 (0.6) |
| 16 | Aritake et al., 2012 [10] | sleep lab (PSG) | nocturnal | 15 (15 / 0) | 22.1 (.7) | \|30\| | 7 (7 / 0) | 47 | |
| 17 | Ikeda et al., 2014 [42] | sleep lab (PSG) | nocturnal | 15 (15 / 0) | 40.5 (6.9) | \|30\| | events | | |

pooling across studies (males = 30%, females = 28%, unknown = 42%). However, when looking at the percentages of females and males self-awakeners within each single paper, several studies reported a higher percentage of females compared to males (n = 5; f/m %: 67/33; 60/40; 75/25; 55/45; 78/22), while only two papers reported the opposite pattern (n = 2; f/m %: 22/78;

22/78), and one paper only included male participants (n = 1; f/m %: 0/100). 5 papers reported the average age in the SA participants and their SD (average reported mean = 38.73 years, average reported SD = 2.38).

## SAQ questionnaire results

SAQ was administered to 164 students from Maastricht University (Maastricht, the Netherlands). Of these, 153 fully completed the testing battery and were retained for analyses (mean age ± SD: 20.62 ± 2.06, 135 F, 18 M, 1 not specified). A descriptive summary of the responses for each item and the proposed score is presented in S1 Table in S1 File.

The maximum score obtainable in the SAQ is 13; to define a possible self-awakener, we propose a cut-off equal to a score of 11 or higher (S1 Table in S1 File). Based on this classification, the current sample included 9/153 *possible* self-awakeners (6%, mean age ± SD: 21.57 ± 1.91, 7F, 2M).

**Self-awakeners vs. non-self-awakeners.** U-statistics and p-values were calculated as the mean results across the 1000 repetitions of the Mann-Whitney performed on the scores of each test in the battery for the SA and nSA groups. Results were non-significant for all the test scores (all p-values > .451; Fig 3).

**Correlations between SAQ and other questionnaires.** Spearman correlations showed significant negative correlations between SAQ and ISI scores ($r_s$ = -.19, p = .020) and between SAQ and PSQI ($r_s$ = -.22, p = .007). However, these results did not survive correction for multiple comparisons (Bonferroni corrected p-value: .05/11 = .005).

**SAQ and circadian preference.** Based on the MEQ scores, participants were divided into a morning (n = 16), an intermediate (n = 100), and an evening (n = 37) groups; their responses

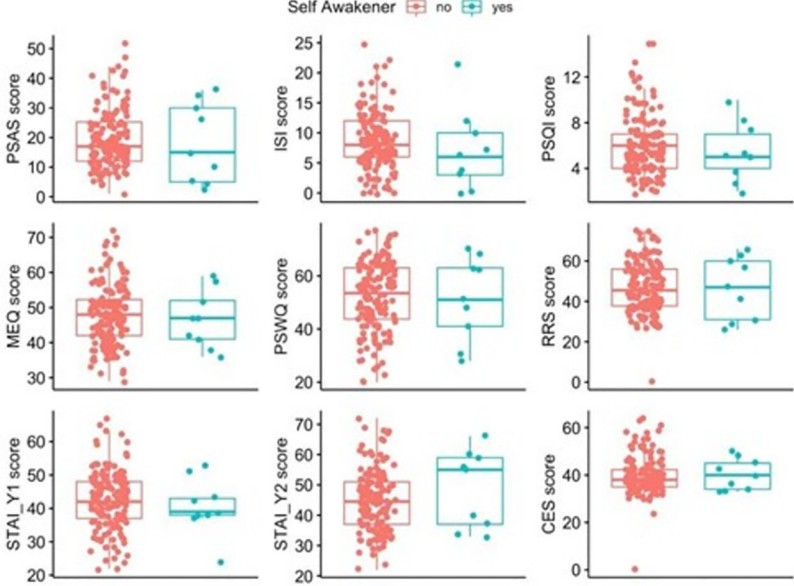

**Fig 3. Differences between Self-Awakeners (n = 9) and non-Self-Awakeners (n = 144) by test scores.** The difference between groups was non-significant for all measures. For all boxplots, each box contains the interquartile range (i.e., the range containing 50% of all observations) and the median value (middle line), while the whiskers represent values extending outside 50% of the observations. Abbreviations: ISI = Insomnia Severity Index; MEQ = Morningness-Eveningness Questionnaire; PSQI = Pittsburgh Sleep Quality Index; PSWQ = Penn State Worry Questionnaire STAI = State-Trait Anxiety Inventory; BDI = Beck Depression Inventory RRS = Ruminative Response Scale; PSAS = Pre-Sleep Arousal Scale; CES = Center for Epidemiologic Studies Depression Scale (see S1 File for details).

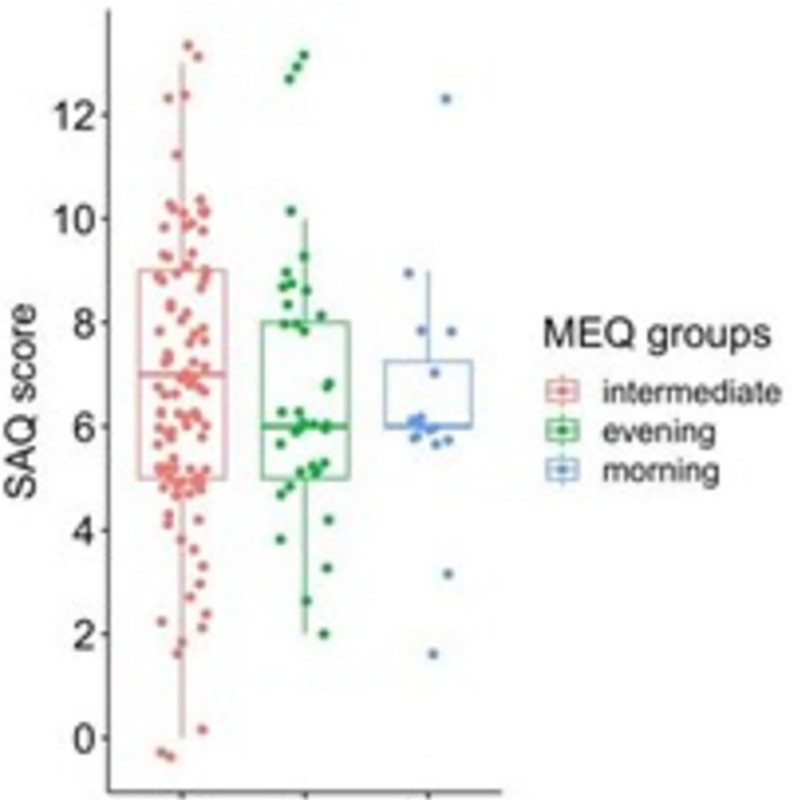

**Fig 4. Distribution of the SAQ scores depending on the respondent's circadian preference group.** Each boxplot contains the interquartile range (i.e., the range containing 50% of all observations) and the median value (middle line), while the whiskers represent values extending outside 50% of the observations.

to each SAQ question were analysed separately (S1 Table in S1 File). The Kruskal-Wallis test showed no significant differences between the three groups (chi-squared = 0.16, df = 2, p = .93; Fig 4).

## Discussion

We reviewed publications investigating the phenomenon of self-awakening up to October 2022. We retrieved 17 papers, written in English, testing a total of 715 participants. Based on these data, we calculated a percentage of self-awakeners corresponding to the 17% of the total pool of participants. The paucity of peer-reviewed papers, as well as the low calculated percentage of self-awakeners, are indicative of self-awakening as a relatively understudied and moderately rare phenomenon. Further, the review revealed that several subjective and objective methods are typically employed in the study of self-awakening.

Although self-awakening is frequently observed in the general population, as reported in the seminal work by Child [7], where 59% of the participants (200 in total) claimed this ability, the definition of the phenomenon is far from uncontroversial or unambiguous. To clarify the phenomenon of interest, we proposed a theoretical disambiguation of this construct (Fig 1), starting from the assumption that self-awakening may represent a privileged model to study the recovery of volition during sleep, a state characterised by a minimal but reversible level of consciousness. Consequently, habitual self-awakening (i.e., waking up every day at the same

time without the aid of external stimuli; e.g., [43]) cannot be considered the topic of our study as circadian factors would be a confounding factor.

In addition to issues related to definition, the current scientific approach to self-awakening is undermined by several methodological problems. For example, objective sleep assessments (e.g., PSG or actigraphy) are rarely used in the literature; yet, because they can reliably certify each participant's wake-up time, they should be the first choice in the study of self-awakenings. Instead, subjective questionnaires and sleep diaries should only be used for screening purposes. Next to objective sleep assessment, one of the biggest methodological issues, acting also as an important source of variability between studies, is represented by the temporal criterion adopted to define a successful self-awakening. Several studies considered different time intervals to define a successful awakening, ranging from ± 5 minutes to ± 40 minutes around the target time. In addition, in some studies, participants were paid based on their accuracy (defined as the difference between target time and actual wake-up time [28, 29]). Importantly, success rate was also influenced when the target time was set close to one's usual habitual wake-up time [32, 34]. All these variables create huge variability across studies.

The neural mechanisms underlying self-awakening are far from being understood. Studies investigating polysomnographic alterations right before self-awakening reported discrepant results. Although it could be possible that the attempt to awake at predetermined time during the night could lighten sleep by reducing slow waves, previous findings do not clearly support this view. Born and collaborators [44] found that time spent in different stages, sleep stages' time distribution, and arousals showed no alteration. Similarly, also Zung and Wilson [28] reported no significant difference in sleep macrostructure between control and experimental nights. Remarkably, a study by Aritake and colleagues [10] investigated the neural correlates of self-awakening through near infrared spectroscopy (NIRS) and polysomnography. They also found no differences in any sleep stages between experimental and control conditions. However, they found an increase in oxyhemoglobin in the right prefrontal cortex and a decrease in delta power 30 minutes before the self-awakening. The increase of oxyhemoglobin in this area is important, since some studies reported its involvement in time recognition [45, 46]. This finding suggests the importance of the activation of brain areas related to time estimation abilities and the disengagement from deep sleep to successfully perform self-awakening.

The second part of this work aimed at developing a questionnaire for screening participants in the general population and for identifying possible self-awakeners, who would then need to be confirmed in an empirical study. According to the proposed scoring method and cut-off score, we identified 9 out of 153 possible self-awakeners. This low percentage (5%) is even more surprising when considering that most participants (98%) answered yes to Question 1 of the SAQ ("Have you ever spontaneously woken up at a desired time, without the help of an alarm clock or other external devices?"). This suggests that most people think they can self-awake or can directly experience it. However, we speculate, the rather low percentage observed when applying the proposed criteria may be indicative of a naive conception of the phenomenon that does not adhere to the proposed definition. Once more, this result underlines the need for a careful disambiguation of the phenomenon, when screening the general population with subjective and retrospective measures. When investigating multiple awakenings before a target time (Question 3: "When you experience these spontaneous awakenings, do you wake up more than once throughout the night?"), 42% of the sample answered positively. This outcome is consistent with a possible deterioration of sleep before a self-awakening as negative sleep changes (increased arousal, wake after sleep onset, and sleep stage 1) are often reported [6]. Yet, the presence of conscious or unconscious awakenings (for example, due to checking the clock) should be excluded. Another important aspect concerns target time: as previously discussed, a person should be able to wake up easily when a target time is set close to their

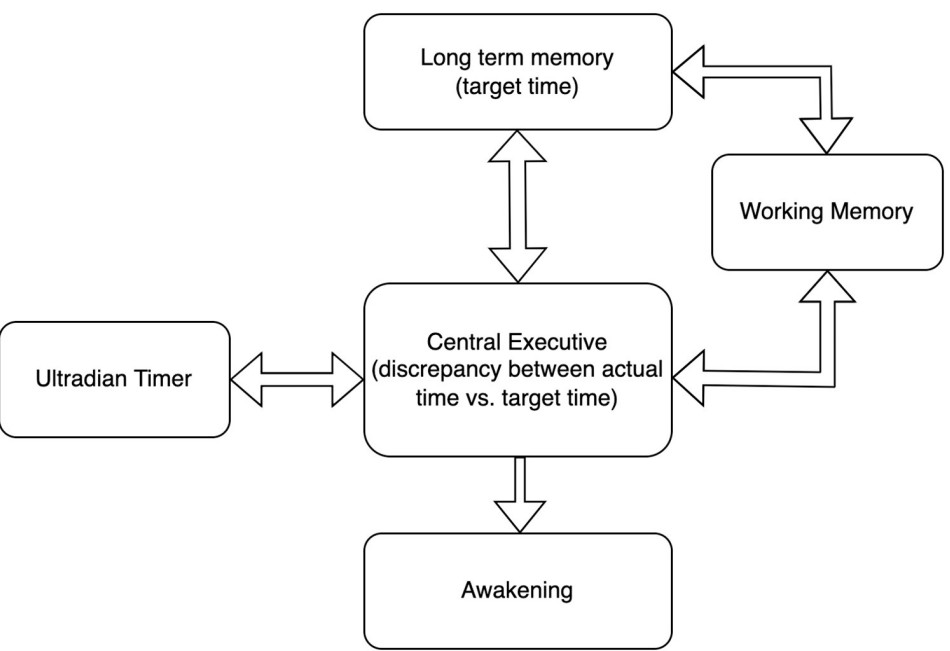

**Fig 5. Proposed schematic model of the cognitive operations potentially involved in self-awakening.**

habitual wake-up time, in particular during the final hours of a sleep period. Question 5 aimed at investigating this aspect ("In your experience, these spontaneous awakenings are more likely to occur..."). Interestingly, 76% of the sample answered that the phenomenon is more likely to occur when waking up at an unusual time (e.g., 3 a.m.). Notably, we observed a weak negative correlation between the score obtained for SAQ, ISI, and PSQI. This relation suggests that self-awakeners (i.e., individuals with a high SAQ score) do not show signs of either insomnia (as evidenced by the low ISI scores) or low sleep quality (as evidenced by the low PSQI scores). This result complies with the definition of self-awakening as an independent phenomenon, unrelated to concomitant sleep disorders. While poor sleep quality or insomnia may also cause spontaneous awakenings, the presence of these disorders should constitute an important exclusion criterion when studying self-awakening.

The investigation of self-awakening is rather relevant, because it might represent an interesting model for studying specific cognitive processes even in states of minimal or absent consciousness. We proposed that three basic cognitive operations may be involved: (i) the encoding and storing of the target time, which must be readily accessible; (ii) the comparison between actual time and target time, which presumes the existence of a psychobiological ultradian timekeeper; (iii) the restoration of volitional control during sleep to wake up (Fig 5).

Although simple, the model proposed in Fig 5 highlights several important points, in particular the lack of a reliable model for time estimation in the minutes-to hours range. This could be explained by the poor knowledge on brain ultradian oscillators (i.e., biorhythm having a period of less than 24 hours) in relation to sleep perception during sleep. Rather, processes involved in the internal timekeeping of circadian rhythms have been extensively studied. These are biologically implemented in a relatively small set of ~10,000 neurons in the suprachiasmatic nucleus the hypothalamus and oscillate with a 24-hour rhythm [47]. Little is known about the timekeepers of sleep (ultradian and not circadian). Some studies directly or indirectly investigated the relationship between time perception and sleep state but with rather

varied results. For example, Aritake-Okada and colleagues [48] showed that time estimation ability during sleep is associated with slow wave sleep. In particular, deep sleep led to an over-estimation of elapsed time. A recent article [49] challenged the common notion that subjective sleep depth, and hence elapsed time, is associated with slow wave sleep, pointing to the relevance of dream-like conscious experiences in perceiving sleep depth. Strikingly, none of these studies directly evaluated the contribution of oscillatory brain activity of time perception during sleep. Future insights into this issue will be fundamental to understand and explain the phenomenon of self-awakening.

Lastly, we put forward some limitations of the current research. First, as stated in the introduction, there are relatively few studies on self-awakening; consequently, the conclusions that can be drawn from our systematic review are preliminary. We sincerely hope that the current manuscript may stimulate further research and lead to a more in-depth understanding of this fascinating phenomenon. Second, the number of self-awakeners compared to non-self-awakeners in our student sample was rather low: Because of the lack of a screening instrument and the relatively low percentage of self-awakeners in the general population, we could not select directly self-awakeners to match non-self-awakeners, nor was it possible for us to test indefinitely. While we partially addressed this issue with our statistical approach, the development of a screening questionnaire–such as the one presented here–may tackle this problem more effectively, by allowing researchers to pre-select and match groups of non- and self-awakeners. Third, our study lacked an objective sleep evaluation to confirm whether the self-awakeners identified by SAQ were indeed capable of spontaneously waking up at a desired time. Objective sleep evaluation by means of PSG is of paramount importance in the study of self-awakening, and we strongly advise future studies to include these measurements.

In conclusion, 17 studies were selected for a review of the literature on self-awakening. These studies reported self-awakening as a relatively rare and understudied phenomenon. However, important methodological differences are present between studies, ranging from the experimental designs to the criteria for defining successful self-awakening and evaluating its accuracy. These differences significantly impact the interpretation of current results. The proposed questionnaire offers first interesting insights on self-awakening and might represent a useful screening tool for studies targeting self-awakening and, more in general, volition during sleep. Future studies employing objective measurement such as EEG, PSG, and actigraphy, with the help of a comprehensive and innovative study design and a clear definition of self-awakening, are needed for a rigorous scientific study of this fascinating phenomenon.

## Supporting information

**S1 File. Additional results, details, and SAQ, Italian version.**
(PDF)

## Author Contributions

**Conceptualization:** Laura Verga, Andrea Galbiati.

**Data curation:** Laura Verga, Giada D'Este, Sara Cassani, Caterina Leitner, Andrea Galbiati.

**Formal analysis:** Laura Verga, Giada D'Este, Sara Cassani, Caterina Leitner, Andrea Galbiati.

**Investigation:** Laura Verga, Andrea Galbiati.

**Methodology:** Laura Verga, Giada D'Este, Sara Cassani, Caterina Leitner, Andrea Galbiati.

**Project administration:** Laura Verga, Andrea Galbiati.

**Resources:** Sonja A. Kotz, Luigi Ferini-Strambi.

**Software:** Laura Verga.

**Supervision:** Sonja A. Kotz, Luigi Ferini-Strambi, Andrea Galbiati.

**Validation:** Andrea Galbiati.

**Visualization:** Laura Verga, Andrea Galbiati.

**Writing – original draft:** Laura Verga, Andrea Galbiati.

**Writing – review & editing:** Laura Verga, Giada D'Este, Sara Cassani, Caterina Leitner, Sonja A. Kotz, Luigi Ferini-Strambi, Andrea Galbiati.

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
