## [Decision Letter · Decision Letter 0]

2 Sep 2022

PONE-D-22-15227Sleeping with time in mind? A literature review and a proposal for a screening questionnaire on self-awakeningPLOS ONE

Dear Dr. Verga,

Thank you for submitting your manuscript to PLOS ONE. After careful consideration, we feel that it has merit but does not fully meet PLOS ONE’s publication criteria as it currently stands. Therefore, we invite you to submit a revised version of the manuscript that addresses the points raised during the review process.

 Please address all comments raised by the reviewer.

We look forward to receiving your revised manuscript.

Kind regards,

Manuel Spitschan

Academic Editor

PLOS ONE

Journal Requirements:

3. Please note that supplementary tables (should remain/ be uploaded) as separate "supporting information" files".

Reviewers' comments:

Reviewer's Responses to Questions

**Comments to the Author**

1. Is the manuscript technically sound, and do the data support the conclusions?

Reviewer #1: Partly

Reviewer #2: Partly

2. Has the statistical analysis been performed appropriately and rigorously? 

Reviewer #1: Yes

Reviewer #2: No

3. Have the authors made all data underlying the findings in their manuscript fully available?

Reviewer #1: No

Reviewer #2: No

4. Is the manuscript presented in an intelligible fashion and written in standard English?

Reviewer #1: Yes

Reviewer #2: Yes

5. Review Comments to the Author

Reviewer #1: Summary:

The paper by Verga et al. titled “Sleeping with time in mind? A literature review and a proposal for a screening questionnaire on self-awakening” investigates the ability to self-awake at a pre-set time without external help, i.e. the ability to end one’s sleep episode voluntarily. The authors provide a systematic literature review following the PRISMA guidelines, summarise the results of the selected studies, present a working definition for “self-awakening” and present their own screening questionnaire they developed based on the literature review. Their questionnaire was designed in English and translated to Italian and administered to 164 students at Maastricht University.

Altogether, the authors find substantial diversity in the 17 reviewed studies in terms of quality, study design and methods. Furthermore, the concept of “self-awakening” has been operationalised and framed differently across studies and the authors present a working definition for this phenomenon. Based on the review and their own study results, the ability to self-awake does not seem to be a common phenomenon albeit this results is very much limited by the small sample sizes across reviewed studies.

The study has several strength which are the adherence to PRISMA guidelines, access to the data, clarification and proposal of the concept behind “self-awakening”, clear language and writing, nice presentation of the results and a succinct summary. Limitations/weaknesses include the missing PRISMA flowchart and no detail on the quality of the reviewed studies despite this being mentioned, low sample size in their own study, raw data only available upon request and in general a relatively superficial handling and discussion of the phenomenon, possible mechanisms, underlying neurobiological facts or discussion of other known facts from related fields, such as time keeping mechanisms when awake.

I recommend major revisions which have to include a more in-depth discussion of the phenomenon including possible mechanisms.

Major issues:

Introduction:

1. The authors seem to make the point that sleep is still viewed as an inactive brain state. This view is outdated and I advise to discuss this more in light of the current evidence concerning sleep architecture, possibly functions of sleep, circadian mechanism/sleep homeostatic components, and cognitive aspects of sleep and during sleep

2. Please add references to the claim that habitual self-awakening, i.e. waking up at the same time every day, is indeed driven by circadian mechanism. I’m unsure whether the evidence is clear on this.

3. In general, more references and in-depth background to the current topic is warranted. For example, what could be the biological differences between habitual self-awakening and self-awakening ability and so on?

Methods:

4. Could the authors please provide information about the program used for statistical analyses, including the version number and used packages if applicable

5. The data are not really openly available, they are only available upon request. Why?

6. Could the authors please clarify what is meant by “NEAR/0” (lines 120-128)

7. It is unclear why each study could obtain a max. of 13 points (line 152). Please clarify.

8. Cohen’s kappa for the inter-rater reliability was rather low, please comment on the difficulties

9. It in unclear how the sampling procedure for their own study was done, nor why they stopped at a certain amount of participants

10. Why was the cut-off for their SAQ at 11; Could the authors provide an explanation?

11. Lines 191-195: More information is needed to understand what was actually carried out: 1 analysis, 9 separate analyses, or something completely different?

12. SAQ “Aim” section: The information given under “aim” in Table 1 points towards “night before a test or important meeting” -> this is confounded by possibly anxiety feelings or feeling nervous; yet the authors suggest that they are probing for self-awakening explicitly without underlying conditions such as stress or anxiety. Could the authors clarify why they think this is still valid?

13. SAQ Q2 and Q5: I think that a neutral answer option should be given here, e.g. “neither” (Q5) and the scale is not adequate in Q2 (every day vs at least 1/week or only on nights before important events). What about other timescales?

Results:

14. Please clarify what is meant by “ps” (line 298) and “rs” (line 301). Multiple p values and r values? If so, maybe giving a range would be more appropriate.

15. Lines 305: It is unclear to me what the MANOVA actually tested; what were the dependent and what the independent variables?

16. Lines 306-314: Could the term “univariate test” be replaced with the actual name of the test that was conducted?

17. Fig 2: The colour-coding legend “self-awakener” is currently 0 and 1 without explanation. I suggest to change to yes/no instead. Also, I strongly recommend to plot the underlying data points, report the sample size per item if not the same across all plots and give an explanation of the errors bars (SD?). The individual tests are only reported with abbreviations and should be explained in the Figure legend.

18. Fig. 3: Similar to Fig. 2, underlying data points to be added, as well as a description of the error bars and sample size.

19. Could it be that in S1 Q5 the 2 numbers (36 vs 117) in the “all” column gut confused?

Discussion:

19. Line 355 talks about the seminal work of Child but does not explain in detail what it was (this was first presented in the Introduction but could be repeated here for clarity)

20. In general, the discussion is nice to read and succinct. I feel though that it is substantially missing a lot of background information, especially on the context of circadian influences, possible mechanisms, time keeping mechanisms that happen outside of sleep, theoretical underpinnings etc. I strongly urge the authors to provide more information on possible mechanisms

21. Lines 416-421: this section needs more additional references and in depth-description of the underlying mechanisms

22. The discussion does not contain a limitation section. Could the authors reflect critically on their work and mention some limitations?

23. The working definition of their concept should be discussed more carefully in this discussion. What does it contain? What does it not contain? What could the underlying mechanisms be? Why is it so difficult to find a common definition, and so on…

#####

Minor issues:

Introduction:

23. Lines 92-96: The authors could expand this section to meet my recommendations above: this is an interesting observation and provides first hints towards possible mechanisms

24. Line 88: 51% of participants - what was the overall sample size?

25. Lines 101-102: It would be nice to discuss the concept of volition. What is it? What are the brain mechanisms involved? How would this be related to certain sleep stages etc?

Methods:

25. Lines 120-121: Unfortunately, the search is a little bit outdated, maybe the authors should consider re-doing this to check if new studies have been published meanwhile?

26. Lines 120-130: Could the authors please present the search terms differently to facilitate reading?i

27. Lines 131: I don’t quite understand the argument concerning language bias: if only English publications were selected anyways, why does this reduced language bias?

28. Line 139: What is meant with “non-experimental”?

29. Line 153: What was the median and the range?

30. SAQ: Please provide more information about the translation. How many people were involved? Was it back translated into English? Was a professional translator involved?

31. Lines 171-176: please provide information about the individual scales and their interpretation. Do higher number mean more severe insomnia etc?

32. Lines 184-189: The big difference in sample sizes between self-awakeners (n=9) and non self-awakeners (n=144) is problematic. Did the authors consider any measures to increase the sample size in general? On the very positive side, they used bootstrapping methods to account for this limitation.

Results:

33. Did the authors of the reviewed studies try to explain underlying mechanisms? If so, please review them when reviewing the results of the reviewed studies.

34. Line 254: Does the term “significantly higher” refer to a statistical test or does it just mean “much higher”?

35. Lines 256-261: I don’t understand how the 8 papers which looked at gender differences found no gender difference but “within each paper’s self-awakeners’ samples, percentages of females and males […] showed a predominance of studies with a higher % of females compared to males”. What is the difference? How do these 2 sentences come together?

36. Table 2 legend: rather long. Instead I propose to explain some of the terms in the column headers or in the cells (e.g. instead of using the abbreviation “E” just write “events”)

37. If using the MEQ and not the Munich Chronotype Questionnaire, I would rather refer to circadian preference and not chronotype. These two are slightly different concepts.

38. Fig. 3: I would suggest to increase the amount of ticks of the y-Axis, e.g. 0,2,4,6,8 etc to facilitate reading the values

Discussion:

39. Line 349: I would suggest to report the actual number of participants, not just more than 700

40. Line 351-352: “This result suggests self-awakening as a relatively understudied […] phenomenon”. Which result suggest this? It reads like the 17% but this does not make sense. Please clarify.

Reviewer #2: This review summarized 17 studies examining self-awakening and proposes a screening questionnaire for self-awakeners. Out of 715 total participants, 125 (17%) reported being self-awakeners, with an average age of 23.24 years and a slight predominance of males compared to females. 9 out of the 153 Dutch students are possible self-awakeners. These results reveal that studies and findings on self-awakening are a relatively rare phenomenon.

Although the topic is intriguing, several serious concerns arise and are listed below:

1.I would delete the part on lucid dreaming lines 62-67.

2.Line 76 can’t it be a learned behavior and/or circadian rhythm issue?

3.Lines 72-78 need references.

4.Line 108 given your proposed taxonomy which type of self-awakening is being studied? This should be specified.

5.Lines 126-128 search terms cover aspects beyond the self-awaking ability. Also, if different languages were allowed in the search, what were the search terms to prevent bias? Please note, Lines 214-215 only mention English papers and abstracts.

6.Line 120-121 an update of the literature is needed.

7.Line 131 was not elaborated on in later sections

8.Line 143 Quality appraisal – the applicability, hence validity, of the tool is very questionable given that it is designed for cohort studies. How many of your study designs were cohort studies?

9.Line 159 Was there any forward/backward translation done? Foremost if this was applied at Maastricht university, was a Dutch version used? Also, what are the psychometric properties of the items selected from the literature, and the SAQ as well? The tools from which the items were selected should be discussed in the Introduction or even Methods. The re-using of items from the literature needs to be based on their psychometric property and not merely face validity.

10.What is the purpose of an Italian version?

11.Line 171-177 was any randomization applied?

12.Line 183 If SAQ contains 5 questions, and each item is scored as a 0 to 3, how can the maximum score be 13? This is misleading, especially when reading Table 1. A justification for the scoring method is needed. A report of several psychometric properties for the SAQ is lacking.

13.How does the response of Q3 “No, just in the morning before the alarm clock” discriminate sufficiently toward self-awaking ability versus habitual self-awakening? For 58% of the sample, this is the answer. Responses like Q3 prime different interpretations throughout your questionnaire, which will affect the validity and reliability of your SAQ.

14.A power calculation of the needed sample size is required because this study is underpowered. The literature estimate is 17% whereas your sample has 6%.

15.The MANOVA is not valid and should be discarded. Lines 308,311, 312 etc , no Bonferroni correction?

16.Information is needed on the validity/reliability of the test battery applied to Dutch students.

17.Is Line 367 not contradictory to your purpose, i.e., creating the SAQ? Namely, is a screening tool not an outcome measure?

Minor

Line 368, and Line 400 sentences need some English correction

Line 382-383, line 405-410 is speculative, and so is Figure 4.

Authors should state the statistical software used for the analyses

6. PLOS authors have the option to publish the peer review history of their article (what does this mean?). If published, this will include your full peer review and any attached files.

Reviewer #1: No

Reviewer #2: No

---

## [Author Response · Author response to Decision Letter 0]

5 Dec 2022

The response to the reviewers has been provided as a separate file with detailed point-by-point replies.

---

## [Decision Letter · Decision Letter 1]

8 Feb 2023

PONE-D-22-15227R1Sleeping with time in mind? A literature review and a proposal for a screening questionnaire on self-awakeningPLOS ONE

Dear Dr. Verga,

Thank you for submitting your manuscript to PLOS ONE. After careful consideration, we feel that it has merit but does not fully meet PLOS ONE’s publication criteria as it currently stands. Therefore, we invite you to submit a revised version of the manuscript that addresses the points raised during the review process.

Please include the points raised by Reviewer 1 in the paper – not just in the rebuttal.

We look forward to receiving your revised manuscript.

Kind regards,

Manuel Spitschan

Academic Editor

PLOS ONE

Journal Requirements:

Additional Editor Comments:

Please include the points raised by Reviewer 1 in the paper – not just in the rebuttal.

Reviewers' comments:

Reviewer's Responses to Questions

**Comments to the Author**

1. If the authors have adequately addressed your comments raised in a previous round of review and you feel that this manuscript is now acceptable for publication, you may indicate that here to bypass the “Comments to the Author” section, enter your conflict of interest statement in the “Confidential to Editor” section, and submit your "Accept" recommendation.

Reviewer #1: (No Response)

2. Is the manuscript technically sound, and do the data support the conclusions?

Reviewer #1: Yes

3. Has the statistical analysis been performed appropriately and rigorously? 

Reviewer #1: Yes

4. Have the authors made all data underlying the findings in their manuscript fully available?

Reviewer #1: Yes

5. Is the manuscript presented in an intelligible fashion and written in standard English?

Reviewer #1: Yes

6. Review Comments to the Author

Reviewer #1: Thank you very much for the careful revision of your article and integrating my suggestions. I recommend publication based on the condition that you simply add some of your answer to my questions also to the document since I believe that these might also help other readers to better understand your article. These include the following:

- the explanation about power analysis and why recruitment stopped (point 9) should also be included in the methods

- maybe also include the description of point 10

- Fig. 2 is very nice, thank you for adopting my suggestions. I still miss the different sample sizes for both self-awakeners and non self-awakeners; even though they don’t change, as you suggest, one has to go back and consult somewhere in the text

- point 20: thanks for adding this information to the discussion, this improves it

- point 28: I guess my point was to specify the study design (i.e. observational study) instead of defining it by what it is NOT, i.e. non-experimental

- point 30: please add this information about translation in the method section

7. PLOS authors have the option to publish the peer review history of their article (what does this mean?). If published, this will include your full peer review and any attached files.

Reviewer #1: No

---

## [Author Response · Author response to Decision Letter 1]

14 Feb 2023

Please see the attached file for the detailed response to reviewers.

---

## [Editor Report · Decision Letter 2]

6 Mar 2023

Sleeping with time in mind? A literature review and a proposal for a screening questionnaire on self-awakening

PONE-D-22-15227R2

Dear Dr. Verga,

We’re pleased to inform you that your manuscript has been judged scientifically suitable for publication and will be formally accepted for publication once it meets all outstanding technical requirements.

Kind regards,

Manuel Spitschan

Academic Editor

PLOS ONE
---

## [Editor Report · Acceptance letter]

13 Mar 2023

PONE-D-22-15227R2 

Sleeping with time in mind? A literature review and a proposal for a screening questionnaire on self-awakening 

Dear Dr. Verga:

I'm pleased to inform you that your manuscript has been deemed suitable for publication in PLOS ONE. Congratulations! Your manuscript is now with our production department. 

Kind regards, 

on behalf of

Dr. Manuel Spitschan 

Academic Editor

PLOS ONE